# Perioperative mortality in low-, middle-, and high-income countries: Protocol for a multi-level meta-regression analysis

**Kevin J. McIntyre**[1,2¤a]*, **Yun-Hee Choi**[1☯], **Ava John-Baptiste**[1,2,3☯¤a], **Daniel J. Lizotte**[1,3☯], **Eunice Y. S. Chan**[2,4☯¤b], **Jessica Moodie**[2¤a], **Saverio Stranges**[1,5,6,7☯‡], **Janet Martin**[1,2☯¤a‡]

**1** Department of Epidemiology & Biostatistics, Western University, London, Ontario, Canada, **2** Department of Anesthesia & Perioperative Medicine, Centre for Medical Evidence Decision Integrity Clinical Impact (MEDICI), Western University, London, Ontario, Canada, **3** Interfaculty Program in Public Health, Western University, London, Ontario, Canada, **4** School of Medicine, The Chinese University of Hong Kong, Shenzhen, Guangdong, P. R. China, **5** Department of Medicine, Western University, London, Ontario, Canada, **6** Department of Family Medicine, Western University, London, Ontario, Canada, **7** Department of Clinical Medicine and Surgery, Federico II University, Naples, Italy

☯ These authors contributed equally to this work.
¤a Current address: Medical Evidence Decision Integrity Clinical Impact, Western University, London, Ontario, Canada
¤b Current address: School of Medicine, The Chinese University of Hong Kong, Shenzhen, China
‡ JM and SS are Joint Senior Authors.
* kmcint67@uwo.ca

**Data Availability Statement:** No datasets were generated or analysed during the current study. All relevant data from this study will be made available upon study completion.

## Abstract

### Background

Surgery is an indispensable component of a functional healthcare system. To date there is limited information regarding how many people die during the perioperative period globally. This study describes a protocol for a systematic review and multilevel meta-regression to evaluate time trends regarding the odds of perioperative mortality among adults undergoing a bellwether surgical procedure while accounting for higher order clustering at the national level.

### Methods

Published studies reporting the number of perioperative deaths from bellwether surgical procedures among adults will be identified from MEDLINE, Embase, Cochrane CENTRAL, LILACS and Global Index Medicus. The primary outcome will be the rate of perioperative mortality across time and the secondary outcome will be investigating cause of death over time as a proportion of overall perioperative mortality. Two reviewers will independently conduct full text screening and extract the data. Disagreements will first be resolved via consensus. If consensus cannot be reached a third reviewer will be included to arbitrate. Due to human resource limitations, a risk of bias appraisal will not be conducted. From the included studies a multilevel meta-regression will be constructed to synthesize the results. This model will conceptualize patients as nested in studies which are in turn nested within countries while taking into account potential confounding variables at all levels.

**Funding:** The authors received no specific funding for this work.

**Competing interests:** The authors have declared that no competing interests exist.

## Discussion

The systematic review and multilevel meta-regression that will be conducted based on this protocol will provide synthesized global evidence regarding the trends of perioperative mortality. This eventual study may help policymakers and other key stakeholders with benchmarking surgical safety initiatives as well as identify key gaps in our current understanding of global perioperative mortality.

## Trial registration

**Systematic review registration:** PROSPERO registration number 429040.

## Introduction

Surgery is an indispensable component of a functional healthcare system. To date there is limited information regarding how many people die during the perioperative period globally. Conditions requiring surgical treatment are highly prevalent worldwide, and unmet need for surgery accounts for approximately one-third of the global burden of disease [1–3]. In 2010, an estimated 16.9 million people died due to conditions requiring surgical care [1].

Due to the widespread need for surgery globally there is also a concurrent need to capture and analyze data regarding surgery and surgical systems at all levels, from local through to international. In 2015 the *Lancet Commission on Global Surgery* proposed global surgery indicators for assessing surgical system capacity to provide timely and safe surgical, anaesthesia and obstetric care [1], and these were updated by the Utstein Consensus in 2019 to include: access, workforce, volume, perioperative mortality, and financial risk protection [4]. Of these 5 global surgery indicators, perioperative mortality rate (POMR) stands alone as the sole indicator measuring clinical outcomes of surgical practice. POMR is defined as "deaths from all causes, before discharge (up to 30 days), in all patients who have received any anaesthesia for a procedure done in an operating theatre, divided by the total number of procedures, per year, expressed as a percentage" [4]. While it is hypothesized that surgery is becoming safer, and there has been some data to indicate that this is the case [5], to date there remains significant uncertainty regarding the actual trends in perioperative mortality especially with regards to the possible differences between high-income countries (HICs) and low-, and middle-income countries (LMICs). Previous work has found estimates generally ranging in the low percentages [5–12] but due to the volume of surgery provided, an estimated 313 million procedures annually [1], the difference in these previous estimates translates to differences in potentially hundreds of thousands or even millions of deaths ever year in absolute numbers.

It is recognized that due to the amount of surgery performed globally as well as the diversity of surgical procedures that proxy measures are needed. *The Lancet* Commission on Global Surgery recommended the use of three surgical procedure groups–Ceasarean section, laparotomy and the treatment of open fractures–which have been labelled the Bellwether procedures to serve this function [1]. Together these procedures serve to proxy surgical systems more broadly (i.e. if an institution can perform these three procedures they likely have the human and infrastructural resources to perform most essential surgery) [1]. Furthermore, Truché et al conducted an analysis using The Brazilian National Healthcare Database to assess the correlation between Bellwether POMR and all-procedure POMR [13]. They found that when the Bellwether procedures were analyzed together their POMR correlated modestly with all-

procedure mortality (r = 0.77) whereas individually the procedures were substantially less correlated with all-procedure mortality indicating that analyzing all three Bellwether procedures simultaneously may be a useful proxy for all-procedure mortality [13].

The primary objective of this analysis will be to examine unadjusted POMR values and time trends in perioperative mortality through a systematic review followed by a multilevel meta-regression to estimate the unadjusted and adjusted odds of perioperative mortality over time in adult patients undergoing bellwether surgical procedures while accounting for the nested structure of the data across low-, middle- and high-income settings. Secondary objectives will be to assess cause specific mortalities, as a proportion of the overall POMR and their individual trends across time.

## Materials and methods

This protocol is registered with PROSPERO, registration number 429040. This protocol follows the PRISMA-P guide.

### Eligibility criteria

All studies, from any country, investigating adults undergoing a bellwether procedure, reporting the number of deaths among participants published since 2014 will be eligible for inclusion.

Due to concerns regarding the volume of surgical research arising from resource restricted settings [14] if there are not enough articles meeting the inclusion criteria from LMICs, this review will extract results from Ng-Kamstra et al's (2018) systematic review investigating POMR in LMICs [15] which meet our inclusion criteria. This will supplement the data provided by LMICs to allow for enough data so that the statistical models can provide estimates of POMR and its time trends in LMICs. This contingency plan has been developed under the philosophy that this review is intended to be global, and so being unable to provide estimates from LMICs would have the effect of restricting the analysis to high-income settings. This is largely contrary to the goals of providing a specific focus on LMICs that the field of global surgery explicitly attempts to do, as well as having the unintentional effect of excluding settings where there is existing evidence that higher perioperative mortality may be occurring [1,16,17].

Studies in any language are eligible for inclusion, but searches will be performed only in English.

Further details on inclusion criteria are provided below:

Types of studies and research designs

Only primary studies will be eligible for inclusion. In this context the term "primary study" is defined as follows: Studies that report on data gathered from individual participants, whether this data is provided at the individual level or aggregated to the study level.

These studies can use either experimental designs, quasi-experimental designs (e.g. difference-in-differences, or interrupted time series) or observational designs that allow for incidence data to be captured (e.g. cohort study designs) with no restrictions on whether they are prospective or retrospective in nature.

Case studies, case-control designs, review articles, opinion articles or other articles otherwise not reporting their own patient data, or which are unable to determine the incidence of perioperative mortality will be excluded.

Articles will be assessed on which surgical procedures were performed. If an article has <5% ineligible procedures (e.g. non-bellwether procedures or concomitant surgery), the

article will be included as the potential risk of bias by excluding the article is deemed to be higher risk than the potential biasing effect of including it.

Studies arising from HICs (as defined by a Human Development Index value of ≥0.700 [18] at the median year of data collection) must have at least 200 patients included in the study to increase the probability that at least one death may be recorded in the study. This requirement will be waived for LMICs due to the anticipated sparsity of data.

Overlapping studies will be excluded. To prevent overlapping datasets, studies reporting on potentially overlapping patient populations will be identified, and the most complete data will be included only once. Studies reporting from the same centre or location, but with non-overlapping timeframes or procedures (e.g. repeated reports on POMR from the same centre, but covering different years and/or procedures) will be included and assigned to their analogous year of data collection.

## Target participant characteristics

This study is focused on adult participants undergoing bellwether surgical procedures. For the purposes of this review, adults will be defined as ≥18 years old. Thus, this systematic review will look to include primary studies that meet the following criteria:

- Articles where at least 50% of the participants are ≥18 years old, i.e. where the median age is 18 years or older or studies where adult data are reported separately from pediatric patients.

- If the median age cannot be determined, then the mean age will be used instead using the same criteria i.e. mean age of the sample must be ≥18 years old to be eligible for inclusion.

- In cases where neither the median nor the mean age of the sample is reported, the authors will determine if the range or distribution of age categories dictate that the median/mean of the sample must mathematically be ≥18 (e.g. the median age category is ≥18 or making the most extreme assumptions regarding the information provided, e.g. assuming all patients in the age category <65 are 0 years old, the mean still would be ≥18). If none of these approaches can determine that the study sample was drawn from adults, then the article will be excluded.

- The article must investigate a "bellwether" procedure. These procedures are defined by the *Lancet Commission on Global Surgery* as Caesarean Section, Laparotomy and Treatment of an Open Fracture [1] and given further specificity by Hanna et al [19].

- Since the objective is to quantify POMR for all patients undergoing each bellwether procedure (i.e. all-comers, mixed risk), studies focused *only* on selected high-risk groups undergoing surgery will not be included in the primary analysis if the complete mixed population data is not provided (e.g. studies focused solely on elderly patients, patients with high baseline risk, undergoing reoperation, frail, malnutritioned or anemic patients).

- The article must provide the number of patients who died in the perioperative period, up to 90 days or "in-hospital" deaths will be considered perioperative with regards to whether the article is eligible for inclusion.

## Information sources

MEDLINE, Embase, Cochrane CENTRAL, LILACS and Global Index Medicus databases will be searched for articles related to bellwether surgeries and mortality.

## Search strategy

The search strategy was constructed by a medical librarian (Jessica Moodie) with systematic searching expertise, and the overall strategy will emulate that provided by Ng-Kamstra et al [15].The search terms will include terms for all bellwether procedures according to the list of procedures provided by Hanna et al, 2020 [19]. The search period will be from January 1, 2014 –present using MEDLINE, Embase, Cochrane CENTRAL, LILACs and Global Index Medicus (an initial search was conducted on September 1–2, 2021 and led to 32,728 references found and uploaded into DistillerSR after an initial deduplication screen). In the case that during the analysis stage, there are insufficient data to establish estimates of POMR and trends in the LMIC income groups, this timeframe will be supplemented by Ng-Kamstra's 2018 meta-analysis investigating POMR in LMICs (2009–2014) [15]. This will provide more information for LMICs which are anticipated to have larger gaps in the evidence [14]. In the unlikely case that there are not enough studies in the HICs cluster, the search will be expanded back to 2009 in line with Ng-Kamstra's work. The studies identified by the search will then be screened sequentially through title/abstract and full-text phases to ensure that they meet the inclusion criteria. The full search strategy can be found in S1 File.

## Screening procedure

All primary articles identified by the search strategy will be compiled first in the reference management software DistillerSR [20] for title/abstract, screening and then transferred to COVIDENCE [21] review management software for full text, screening before being transferred to REDCap for data extraction and database creation.

Studies excluded from the review will still be maintained in a separate file for record keeping and transparency purposes. Excluded studies will be documented in a PRISMA flow chart. The title and abstract screening stage will be conducted by a single author, and the full text review will be conducted by two independent reviewers, with conflicts being resolved by consensus. In the case that consensus cannot be reached, a third reviewer will be included to arbitrate.

Title/abstract screening will be conducted by a single reviewer in the DistillerSR review management software. This software uses artificial intelligence to reorganize the references to be screened so that those most likely to be included in the review are consistently brought to the front for review. With this software we will screen articles until we have included at least 90% of predicted relevant references or reached an inclusion rate <5%, as measured through the DistillerSR software. This inclusion rate is calculated by dividing the number of included references by the total number of references in the four most recent blocks of 200 studies. For further information regarding the validity of using artificial intelligence tools in screening for systematic reviews we refer readers to Hamel et al, 2020 and Burns et al, 2021 [22,23].

Once either 90% of predicted relevant references have been found or the inclusion rate reaches <5%, an automated search will be conducted through the remaining studies. At this point all references meeting the inclusion criteria after title/abstract screening will be transferred to the COVIDENCE platform for full text screening. Two reviewers will independently review each reference and then independently extract the data from every reference included in the review. Articles that are excluded by reviewers for different reasons will be assessed via a hierarchy of reasons (see appendix for hierarchy).

A final phase will specifically apply to studies utilizing national, or population level, database data as well as studies utilizing data from multiple countries. Articles which used population-based data will be tagged at the full text screening phase and will be compared against each other after data extraction. These references will be compared based on procedures

conducted, years of data collected and country the procedures were performed in. Finally, only those studies which maximize the number of patients while minimizing the number of likely duplicate patients will be included for analysis. The purpose of this is to ensure that patients are not counted more than once for the same procedure during the meta-analysis as this will inappropriately reduce the variance estimates of the models and possibly bias the point estimates. Articles using data from multiple countries will be identified, and the authors of these studies contacted. We will attempt to gather country-specific data from these studies so that HDI status can be accurately assigned to the participants undergoing surgery and those who experienced perioperative mortality. If this data cannot be collected via contacting the authors, then the study will be excluded. All references not included due to these procedures will be provided in the supplemental files to ensure transparency, reproducibility and allow for independent sensitivity analyses.

## Data extraction

Included studies will have their data extracted and compiled to create a database on POMR and cause-specific POMR. This data extraction will be conducted using a standardized data extraction form that will be created in the REDCap data collection software and piloted using the development mode before use for actual extraction. The data collection instrument will be created as a repeat instrument in the software allowing for each reviewer to extract the data independently and then a final harmonized version to be exported for data analysis.

Data to be extracted consists of an electronic link to the study, title of the article, last name of first author, year of publication, start and end date of data collection, type of study design, whether the data was collected prospectively or retrospectively, name of institutions contributing data, country where the surgeries took place, HDI value of the country at the median year the study collected data (or closest available year), type of surgery, sample size, number of perioperative deaths, proportion of elective surgeries, proportion of patients of each sex, median American Society of Anesthesiologists (ASA) physical score (if available–mean if not), proportion of COVID-in infected patients, level of hospital (if possible), follow-up length, whether the institution was located in an urban or rural setting, whether the study was solely investigating cancers, and specific causes of death. From these variables, derived variables such as median year of data collection and POMR can be calculated.

The proposed data extraction collection form can be found in S4 File.

## Risk of bias assessment

Due to the large scope of the project and the lack of tools to evaluate the types of study designs included in this meta-analysis, risk of bias assessment will not be formally done. However, limitations in the studies will be noted generally, including selection bias, performance bias, attrition bias, and included in the discussion. As such the strength of the body of evidence will be assumed to be weak in place of conducting a formal assessment of the strength of the evidence (e.g. GRADE).

## Data synthesis

The primary purpose of this study is to examine time trends in perioperative mortality, both overall and stratified by bellwether type, using existing research to construct a systematic review and multilevel meta-regressions allowing for the estimation of how the odds (along with 95% CIs) of perioperative mortality are changing over the course of time while accounting for the hierarchical structure of the data. This will be done alongside determining POMR at different time-points. To calculate these POMRs studies will be aggregated within each

corresponding decade (data permitting) and unadjusted POMR will be calculated by summing the number of deaths that occurred and dividing this value by the summed number of procedures conducted. The secondary objective is to assess cause specific mortality time trends as a proportion of overall POMR to discover which causes of death may be driving perioperative mortality more broadly.

A typical meta-analysis is a specialized case of a two-tiered multilevel model where participants are nested within studies to account for heterogeneity both within and between the studies [24–31]. This proposed study will extend a random effects model by additionally accounting for heterogeneity at the country level, adding a third level into this multilevel framework. This proposed framework will model participants as nested within studies which are in turn nested within countries. These specifics can be applied to the general framework as visualized by Harrer et al. in Chapter 10 of their book, *Doing Meta-Analysis in R*: *A Hands-on Guide* [26]. An additional advantage of using a multilevel modeling technique for this analysis is that it inherently accounts for the expected interaction between year of surgery and HDI. This interaction is theoretically plausible and could be an important aspect of trends in global POMR. Hezam (2020) [32] conducted two systematic reviews and meta-analyses using multiple bivariate meta-regressions to show that patterns of perioperative maternal mortality ratios across both HDI and year. However, no model was created that accounted for both these factors simultaneously, nor any potential interaction between them. Thus, by utilizing the proposed methods in this protocol this study plans to account for this possibility.

Under this conceptualization of meta-analysis as a multilevel model and the potential extension via addition of higher levels of clustering, many of the existing features of meta-analysis take on a new interpretation. Here, studies are regarded as clusters sampled from the underlying patient population which themselves are clustered within the nations that they are drawn from. As such, traditional measures used to explore the consistency of data in meta-analysis such as $I^2$, or Kendall's tau become difficult to interpret because these metrics become ways of assessing variance in level two of the model. Thus, provided there are enough primary articles included in the review to meet the typical assumptions necessary for multilevel regression analysis (e.g. large enough sample size, and enough clusters) quantitative synthesis will be conducted which will include variance estimates at every level of the model. In the unlikely event that these assumptions are not met, then a qualitative analysis will be conducted in addition to an Evidence Gap Map (EGM). This analysis will report narratively on themes to provide policymakers and future researchers information on time trends in overall POMR and cause-specific POMR among adults undergoing bellwether surgical procedures in low-, middle-, and high-income countries.

Another advantage of using a generalized linear mixed model (GLMM) based approach is that they can effectively handle studies that provide proportions of 0% rather than having to rely on the use of continuity corrections [24,25,33]. Since POMR is a rare event, and the risk is not evenly distributed among surgeries, there is a high likelihood that several studies will include no perioperative mortalities. This issue is negated by utilizing this type of model.

Additionally, extending the model to include covariates is straightforward in the GLMM space. This is another advantage for this study to help reduce confounding and account for unexplained variation between studies [30]. In addition to investigating the effects of HDI and time, on POMR other variables that are proposed to be important confounders of this relationship can also be included in the GLMM. These covariates will include median or average age of the participants in each study, proportion of elective surgeries in each study, median or mean ASA score of each study, proportion of female sex in each study, proportion of COVID positive patients in each study, hospital level (primary, secondary, tertiary or quaternary) and whether the facility was located in an urban or rural setting. Several models will be analyzed

including an overall model, as well as models stratified by bellwether procedure. All models will investigate whether a time increase is increasing or decreasing the odds of perioperative mortality in studies as assessed by the odds ratio with corresponding 95% confidence intervals. Since time will be modeled as a continuous variable, a significant trend will be determined if the confidence intervals for the beta coefficient related to the time variable does not encompass the null value of one while accounting for other variables that could confound the results as mentioned above. These analyses will ultimately be conducted using a traditional meta-analysis approach as well as the three-level approach so that the potential effect of extending the model to include clustering at the national level can be assessed. All analyses will be conducted using the R statistical analysis software.

The primary articles will also be used to populate an Evidence Gap Map (EGM), to provide a visual guide to the degree that this data exists and in what countries.

## Secondary objectives and analyses

The secondary objective is to investigate cause-specific mortality in an attempt to understand the causes driving POMR. Data on causes will first be presented as overall proportions aggregated by type of surgery, time and HDI category wherever possible. Sankey diagrams will be used to better visualize changes over time in these proportions to provide a clearer picture of the impact of the causes on perioperative mortality.

## Outcomes

The primary outcome to be extracted from the articles is the number of perioperative deaths reported. POMR will then be calculated by dividing this number by the number of procedures performed in the study.

The secondary outcome involves investigating causes of death. To extract this data a text field will first be utilized to capture the reported causes. Depending on the quality of the data extracted causes may have to be aggregated into larger categories. This is due to the lack of a consistent, global, cause of death classification system and the likely issue of fewer studies reporting this specific information.

## Planned additional subgroup and sensitivity analyses

Due to the scope of this proposed project and the foreseeable difficulties in accounting for many aspects that may have an impact on the results of the analyses, several additional analyses, sensitivity analyses and subgroup analyses will be conducted.

## Additional analyses

- An analysis to assess the risk of publication bias across studies using funnel plots and Egger's test.

## Subgroup analyses

- Analyses stratified by bellwether procedure type (i.e. laparotomy specific, cesarian section specific and treatment of open fracture specific models will be constructed).

- Analyses stratified by HDI group (i.e. low, middle-, and high-income HDI groups) to ensure that the anticipated volume of data from high-income settings does not bias the results and that there are estimates provided for all three income setting categories.

- A subgroup analysis comparing observational designs to RCTs.

## Sensitivity analyses

- A sensitivity analysis removing studies from LMICs that did not meet the 200-person sample size restriction will be conducted to assess if this differential inclusion restriction affected the results.

- A sensitivity analysis to assess whether assumptions regarding missing data of baseline covariates affected POMR estimates (e.g. multiple imputation versus complete case analysis).

- To assess the effect of the GLMM model a sensitivity analysis will be conducted using a second method to handle data presenting extreme values. This analysis will be identical to the initial model only instead of using the GLMM method directly on the data extracted from the individual studies, the data will first be analyzed using an empty Bayesian model with a non-informative conjugate prior. This will transform the values of the original data slightly serving as a sort of principled continuity correction as it will transform proportions of 0% to proportions approaching this null value. The results from this analysis will then be used as the data to perform the GLMM. The estimated odds of perioperative mortality using this technique will then be assessed against the initial analyses to investigate what effect these methods of analyses may have had on the results.

- Due to the potential prognosis differences, and the heterogeneity of various cancers in both surgical methods and mortality risks, an analysis excluding studies investigating cancers will be conducted.

- A subgroup analysis assessing the impact of various follow-up times cut-offs for the definition of POMR (e.g. in-hospital, 30-day, 60-day, 90-day).

## Discussion

Surgery is an indispensable aspect of any functional healthcare system. Among the proposed metrics to capture data regarding the progress of establishing universal access to safe and affordable surgery POMR stands alone as the sole measure of patient safety. As such it is important for health systems to understand both the current rate of perioperative mortality as well as where such a rate fits into the broader historical trends.

In addition to the importance of determining the trends of perioperative mortality globally, an important methodological strength of this proposed systematic review and multilevel meta-regression analyses is that it will provide better estimates regarding the variance surrounding the individual country-level estimates as to the trend regarding perioperative mortality. Accurately capturing the uncertainty surrounding the estimates provided is an important development especially when considering the higher order clustering that can occur due to country level factors such as national policies. Additionally, this method will provide estimates that are not biased downwards due to the anticipated disparity in the volume of data arising from high-income countries which are hypothesized to have a lower perioperative mortality rate than more resource constrained settings. This can then be compared to analyses stratified by HDI income groups which provide an alternative method for assessing POMR in different resource settings.

While this protocol has many strengths there are several limitations that must also be mentioned. Notably, abdominal aortic aneurysm repair, esophagectomy, percutaneous procedures and endovascular procedures will not be considered bellwether procedures for the purposes of

this review. This is because abdominal aortic aneurysm repair is traditionally considered more of a vascular procedure rather than a general surgery; esophagectomy can include thoracic involvement; and percutaneous and endovascular procedures are typically not considered surgery as they are often conducted in a procedure room rather than an operating room. Additionally, studies conducted in military hospital settings will be excluded due to the fact that they do not represent either the patient population of the nation where they are conducted in nor the patient population of the nationality of the military itself.

Since we have insufficient resources to capture unpublished and grey literature, this may increase the possibility of publication bias. Furthermore, with the limited human resources and lack of validated instruments for bias assessment of this type of study design, we have deprioritized risk of bias assessment, and the quality of evidence may be regarded as at a high risk of bias. However, this analysis is intended as a first attempt to provide large scale estimates regarding POMR and should not be viewed as definitive. Instead, it should be viewed as a starting point from which future work can be benchmarked.

By investigating the published literature and providing a synthesized estimate of surgical mortality, as well as the existing trends regarding its incidence this project could help create an estimate of where we need to be focusing to help make surgery safer. Additionally, the data resulting from the eventual meta-analysis may be useful as a benchmark from which future system-level quality initiatives could use as a baseline in assessing future surgical performance. The findings from this project may help policymakers and other stakeholders assess the quality of surgical systems in their jurisdictions as well as plan for future surgical system needs.

## Supporting information

**S1 File. Search strategy.**
(DOCX)

**S2 File. Screening protocol.**
(DOCX)

**S3 File. Exclusion criteria hierarchy.**
(DOCX)

**S4 File. Data extraction form.**
(PDF)

**S5 File. PRISMA-P 2015 checklist.**
(DOCX)

## Author Contributions

**Conceptualization:** Kevin J. McIntyre, Saverio Stranges, Janet Martin.

**Methodology:** Kevin J. McIntyre, Yun-Hee Choi, Ava John-Baptiste, Daniel J. Lizotte, Eunice Y. S. Chan, Jessica Moodie, Janet Martin.

**Project administration:** Jessica Moodie.

**Resources:** Eunice Y. S. Chan, Saverio Stranges, Janet Martin.

**Supervision:** Saverio Stranges, Janet Martin.

**Writing – original draft:** Kevin J. McIntyre.

**Writing – review & editing:** Yun-Hee Choi, Ava John-Baptiste, Daniel J. Lizotte, Eunice Y. S. Chan, Jessica Moodie, Saverio Stranges, Janet Martin.

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
