## [Decision Letter · Decision Letter 0]

14 Dec 2023

PONE-D-23-20961Perioperative mortality in low-, middle-, and high-income countries: Protocol for a multi-level meta-regression analysisPLOS ONE

Dear Dr. McIntyre,

Thank you for submitting your manuscript to PLOS ONE. After careful consideration, we feel that it has merit but does not fully meet PLOS ONE’s publication criteria as it currently stands. Therefore, we invite you to submit a revised version of the manuscript that addresses the points raised during the review process.

Please revise.

We look forward to receiving your revised manuscript.

Kind regards,

Academic Editor

PLOS ONE

Journal Requirements:

Reviewers' comments:

Reviewer's Responses to Questions

**Comments to the Author**

1. Does the manuscript provide a valid rationale for the proposed study, with clearly identified and justified research questions?

Reviewer #1: Yes

Reviewer #2: Yes

2. Is the protocol technically sound and planned in a manner that will lead to a meaningful outcome and allow testing the stated hypotheses?

Reviewer #1: Yes

Reviewer #2: Yes

3. Is the methodology feasible and described in sufficient detail to allow the work to be replicable?

Reviewer #1: Yes

Reviewer #2: Yes

4. Have the authors described where all data underlying the findings will be made available when the study is complete?

Reviewer #1: Yes

Reviewer #2: Yes

5. Is the manuscript presented in an intelligible fashion and written in standard English?

Reviewer #1: Yes

Reviewer #2: Yes

6. Review Comments to the Author

You may also provide optional suggestions and comments to authors that they might find helpful in planning their study.

Reviewer #1: The authors have created a beautifully written study protocol to examine the perioperative mortality in low, middle, and high income countries using a meta-regression analysis. The authors want to determine the POMR amongst bellweather procedures.

My first concern with this protocol is the use of DistillerSR as this is an artificial intelligence software that will organize the references and only include the relevant studies. Has this software ever been verified or utilized for a meta-analysis? How does the reviewer double check DistillerSR's work and are they able to identify any articles that the software may have missed that could be relevant to the study. How can the software or the authors determine that articles have <5% ineligible procedures?

The protocol to perform a meta-analysis is not a novel protocol, even if this is a meta-analysis of a "grand scale". The authors are reaching too broadly and need to narrow their topic down to just one specific surgery or surgical sub-specialty instead of trying to perform a meta-analysis across multiple countries/surgical specialties/procedures to gain any meaningful data.

What are the authors' main objective with this study? To determine if high income countries have better POMR than lower income countries? How would that improve the healthcare field clinically or in day-to-day processes? It has been well-established that higher income countries have improved health care outcomes; so, what are the authors hoping to prove with this study?

Reviewer #2: Dear Authors,

It is a pleasure to read your protocol. Assessing perioperative mortality with its causes and time trends is a of great importance to provide recommendations on how to reduce or avoid it. However, this peer review targets mainly the systematic review methodology, which is my relvant area of experience.

It is a well-written protocol with clear justifications of the proposed methods.

There are only a pair of concerns to clarify to the readers:

- Clarify what you mean with the sentence line 130-131 "given the scarcity of published research in these settings" given that the number of included studies of the Ng-Kamstra et al study is 985 studies?

- Will you pay attention in your subgroup analysis to the impact of the potential differences (i.e., technological or medical advancement) of the healthcare systems from 2009 till now?

7. PLOS authors have the option to publish the peer review history of their article (what does this mean?). If published, this will include your full peer review and any attached files.

Reviewer #1: No

Reviewer #2: No

---

## [Author Response · Author response to Decision Letter 0]

25 Jan 2024

Dear Dr Chen,

We would like to thank you and the reviewers for your time in reading our manuscript and the thought that has clearly gone into the reviewer comments. Below we offer a point-by-point response to these comments.

Reviewer #1: My first concern with this protocol is the use of DistillerSR as this is an artificial intelligence software that will organize the references and only include the relevant studies. Has this software ever been verified or utilized for a meta-analysis? 

Response: Yes, there has been two validation study conducted by Hamel et al in 2020 and Burns et al in 2021 (1,2). Furthermore, there have been at least three other protocols published using DistillerSR in the past several years (3–5). 

We believe that Reviewer #1 certainly has a right to be concerned about the implementation of artificial intelligence methods in systematic review screening, especially given the lack of detail regarding the validity of this technique in the initial manuscript. We have incorporated a sentence to provide readers with the context of this existing validation work (lines 246-248).

Reviewer #1: How does the reviewer double check DistillerSR's work and are they able to identify any articles that the software may have missed that could be relevant to the study. 

Response: Unfortunately, there is no way to double check the work to ensure that all articles have been captured without manually screening through every article which would make using an AI tool to assist moot. We are proposing the use of artificial intelligence software in this proposed systematic review as an attempt to solve this dilemma. Our strategy will minimize the risk to validity through using DistillerSR which constantly re-organizes the unseen references into a descending probability of inclusion order rather than the random order that they would otherwise be in. A human will screen these references until we reach the prespecified threshold where we will run a fully automated search on the remaining references to minimize the risk of missing studies.

Reviewer #1: How can the software or the authors determine that articles have <5% ineligible procedures?

Response: During level one screening we manually will exclude studies where it is clear that >5% of the procedures are ineligible. Otherwise, this will be done at the full text screening stage. Using Hanna et al’s list of bellwether procedures as a guide, we will determine the proportion of procedures ineligible by dividing the number of ineligible procedures by the total number of procedures (6). This is also an attempt to mitigate bias due to unforeseen instances where a small proportion of ineligible procedures may otherwise force us to discard an entire study filled with otherwise valuable information. Given the nature of potential for bias either way we think that the risk of bias by including these studies is less than the potential risk of bias that would be incurred by excluding them.

Reviewer #1: The protocol to perform a meta-analysis is not a novel protocol, even if this is a meta-analysis of a "grand scale". The authors are reaching too broadly and need to narrow their topic down to just one specific surgery or surgical sub-specialty instead of trying to perform a meta-analysis across multiple countries/surgical specialties/procedures to gain any meaningful data.

Response: We appreciate the concern about the breadth of this proposed project as it certainly is large. To clarify the importance of investigating Bellwether procedures, together they are an important proxy in global surgery for assessing surgery as a whole as proposed by The Lancet Commission on Global Surgery and further demonstrated by Truché et al (7,8). With regard to the novelty of this review, to our knowledge this is the first systematic review to assess POMR in all three bellwether procedures in high-, middle- and low-income countries as well as providing estimates of time trends in POMR. Additionally, to our knowledge this is the first systematic review and meta-analysis to use the multilevel meta-regression methodology in the global surgery field. We believe that the knowledge generated from this review will be helpful for policy makers with regard to surgical system planning. We have added a sentence explaining the importance of Bellwether procedures (lines 115-127)

Reviewer #1: What are the authors' main objective with this study? To determine if high income countries have better POMR than lower income countries? How would that improve the healthcare field clinically or in day-to-day processes? It has been well-established that higher income countries have improved health care outcomes; so, what are the authors hoping to prove with this study?

Response: The main objective with this study is to investigate trends in POMR. This can be conceptualized as regressing POMR on median year of data collection, which will provide information regarding what the current trends are globally. 

We absolutely agree that it is well-established that HICs have better health outcomes regarding POMR as demonstrated by Bainbridge et al (9). The overall models (not stratified by income setting) will use HDI as a continuous covariate to account for differences in country income. A secondary outcome of interest is what the trends of POMR are within income settings, i.e. after stratifying on HDI group. The purpose of these analyses is to look at the trends of POMR within strata rather than comparing between strata. This may show different trends in POMR in different income settings. We have also incorporated a sentence discussing the potential impact of this project (lines 448-449)

Reviewer #2: There are only a pair of concerns to clarify to the readers:

- Clarify what you mean with the sentence line 130-131 "given the scarcity of published research in these settings" given that the number of included studies of the Ng-Kamstra et al study is 985 studies?

Response: This is a very good question by Reviewer #2. This was a mistake from an earlier draft that was wrongly worded. We are worried that given the well-documented disparity in the volume of published research output between high income settings and LMICs, there will not be a substantial number of articles arising from resource restricted settings that also meet the strict inclusion criteria to be included in this review(10). The inclusion of studies from Ng-Kamstra et al’s systematic review is a contingency plan in the case that we do not find enough articles from LMIC settings to conduct statistical analyses which would have the effect of restricting our analyses to HICs only. We have updated the manuscript to clarify this contingency plan (lines 142-152 and 220-222). 

Reviewer #2: Will you pay attention in your subgroup analysis to the impact of the potential differences (i.e., technological or medical advancement) of the healthcare systems from 2009 till now?

Response: This is a very insightful question. We believe that this is likely, at least in part, the mechanism by which trends in POMR may be being impacted over time. In its truest sense we would not expect the median year of data collection to influence POMR itself. If the only thing that changed was a one-year increase in median year of data collection, we do not see a theoretical reason why this alone would impact POMR directly. The mechanism in which time affects POMR is through the impact of such factors such as technological, medical, educational, or other policy advancements and/or changes.

Kind regards,

Kevin McIntyre

References

1. Hamel C, Kelly SE, Thavorn K, Rice DB, Wells GA, Hutton B. An evaluation of DistillerSR’s machine learning-based prioritization tool for title/abstract screening – impact on reviewer-relevant outcomes. BMC Med Res Methodol. 2020 Dec;20(1):256. 

2. Burns JK, Etherington C, Cheng‐Boivin O, Boet S. Using an artificial intelligence tool can be as accurate as human assessors in level one screening for a systematic review. Health Info Libraries J. 2021 Nov 18;1–13. 

3. Cao J, Chang-Kit B, Katsnelson G, Far PM, Uleryk E, Ogunbameru A, et al. Protocol for a systematic review and meta-analysis of the diagnostic accuracy of artificial intelligence for grading of ophthalmology imaging modalities. Diagn Progn Res. 2022 Dec;6(1):15. 

4. Hester K, Kirrane E, Anderson T, Kulikowski N, Simmons JE, Lehmann DM. Environmental exposure to metals and the development of tauopathies, synucleinopathies, and TDP-43 proteinopathies: A systematic evidence map protocol. Environment International. 2022 Nov;169:107528. 

5. Verret M, Lam NH, Fergusson DA, G Nicholls S, Turgeon AF, McIsaac DI, et al. Intraoperative pharmacologic opioid minimisation strategies and patient-centred outcomes after surgery: a scoping review protocol. BMJ Open. 2023 Mar;13(3):e070748. 

6. Hanna JS, Herrera-Almario GE, Pinilla-Roncancio M, Tulloch D, Valencia SA, Sabatino ME, et al. Use of the six core surgical indicators from the Lancet Commission on Global Surgery in Colombia: a situational analysis. The Lancet Global Health. 2020 May;8(5):e699–710. 

7. Truché P, Shoman H, Reddy CL, Jumbam DT, Ashby J, Mazhiqi A, et al. Globalization of national surgical, obstetric and anesthesia plans: the critical link between health policy and action in global surgery. Global Health. 2020 Dec;16(1):1. 

8. Meara JG, Leather AJM, Hagander L, Alkire BC, Alonso N, Ameh EA, et al. Global Surgery 2030: evidence and solutions for achieving health, welfare, and economic development. 2015;386:56. 

9. Bainbridge D, Martin J, Arango M, Cheng D. Perioperative and anaesthetic-related mortality in developed and developing countries: a systematic review and meta-analysis. The Lancet. 2012 Sep;380(9847):1075–81. 

10. Sullivan R. The State of Research. North American Launch of the Lancet Commission on Global Surgery. North American Launch of the Lancet Commission on Global Surgery.; 2015; Boston.

---

## [Decision Letter · Decision Letter 1]

17 Mar 2024

PONE-D-23-20961R1Perioperative mortality in low-, middle-, and high-income countries: Protocol for a multi-level meta-regression analysisPLOS ONE

Dear Dr. McIntyre,

Thank you for submitting your manuscript to PLOS ONE. After careful consideration, we feel that it has merit but does not fully meet PLOS ONE’s publication criteria as it currently stands. Therefore, we invite you to submit a revised version of the manuscript that addresses the points raised during the review process.

Please revise.Please submit your revised manuscript by May 01 2024 11:59PM. If you will need more time than this to complete your revisions, please reply to this message or contact the journal office at plosone@plos.org. Please include the following items when submitting your revised manuscript:A rebuttal letter that responds to each point raised by the academic editor and reviewer(s). You should upload this letter as a separate file labeled 'Response to Reviewers'.A marked-up copy of your manuscript that highlights changes made to the original version. You should upload this as a separate file labeled 'Revised Manuscript with Track Changes'.An unmarked version of your revised paper without tracked changes. You should upload this as a separate file labeled 'Manuscript'.If applicable, we recommend that you deposit your laboratory protocols in protocols.io to enhance the reproducibility of your results. Protocols.io assigns your protocol its own identifier (DOI) so that it can be cited independently in the future. For instructions see: https://journals.plos.org/plosone/s/submission-guidelines#loc-laboratory-protocols. Additionally, PLOS ONE offers an option for publishing peer-reviewed Lab Protocol articles, which describe protocols hosted on protocols.io. Read more information on sharing protocols at https://plos.org/protocols?utm_medium=editorial-email&utm_source=authorletters&utm_campaign=protocols.

We look forward to receiving your revised manuscript.

Kind regards,

Academic Editor

PLOS ONE

Journal Requirements:

Reviewers' comments:

Reviewer's Responses to Questions

**Comments to the Author**

1. Does the manuscript provide a valid rationale for the proposed study, with clearly identified and justified research questions?

Reviewer #1: Yes

Reviewer #3: Yes

2. Is the protocol technically sound and planned in a manner that will lead to a meaningful outcome and allow testing the stated hypotheses?

Reviewer #1: Yes

Reviewer #3: Yes

3. Is the methodology feasible and described in sufficient detail to allow the work to be replicable?

Reviewer #1: Yes

Reviewer #3: Yes

4. Have the authors described where all data underlying the findings will be made available when the study is complete?

Reviewer #1: Yes

Reviewer #3: Yes

5. Is the manuscript presented in an intelligible fashion and written in standard English?

Reviewer #1: Yes

Reviewer #3: Yes

6. Review Comments to the Author

You may also provide optional suggestions and comments to authors that they might find helpful in planning their study.

Reviewer #1: The authors have addressed most of my concerns.

I would still recommend the authors also address/control for the publication that occurs between high vs. low income countries. The amount of data available regarding high income countries that have the funds to publish their data will always outweigh those of lower income countries will skew the data and the authors should address these issues and control for that if possible. Would DistillerSR be able to account for this bias?

Reviewer #3: The revised manuscript was good written and all reviewrs' comments were addresse in the manuscript.

7. PLOS authors have the option to publish the peer review history of their article (what does this mean?). If published, this will include your full peer review and any attached files.

Reviewer #1: No

Reviewer #3: No

---

## [Author Response · Author response to Decision Letter 1]

25 Mar 2024

Dear Dr Chen,

We would like to again thank you and the reviewers for reading our manuscript and identifying a key issue in global health meta-analysis. Below we provide a response to the reviewer’s comment. 

Reviewer #1: I would still recommend the authors also address/control for the publication that occurs between high vs. low income countries. The amount of data available regarding high income countries that have the funds to publish their data will always outweigh those of lower income countries will skew the data and the authors should address these issues and control for that if possible. Would DistillerSR be able to account for this bias?

Response: The issue of high-income countries skewing the results and masking perioperative mortality in low-income settings is a concern of ours as well. Unfortunately, DistillerSR will not be able to account for this issue. However, the problem that you have raised is the rationale for why we are conducting a multilevel analysis. This statistical technique extends the typical random effects meta-analysis to include the clustering variable of countries. By including this level, the analysis will account for clustering patterns within and between countries rather than just taking a weighted average of all the studies based upon the inverse of their variance thereby accounting for this bias. Furthermore, in the meta-regression analyses HDI will be included as a covariate in the model which should also help statistically account for this influence. In addition to these methods, we have also added an analysis of POMR stratified by income setting (ie low, middle, and high HDI groups) to ensure the data from high-income countries are not skewing the results that would be found in low resource settings using more traditional methods. Lastly, we have added a sentence to the discussion to further inform readers of this benefit regarding the use of the multilevel meta-regression analysis (lines 428-432).

Kind regards,

Kevin McIntyre

---

## [Decision Letter · Decision Letter 2]

16 Apr 2024

Perioperative mortality in low-, middle-, and high-income countries: Protocol for a multi-level meta-regression analysis

PONE-D-23-20961R2

Dear Dr. McIntyre,

We’re pleased to inform you that your manuscript has been judged scientifically suitable for publication and will be formally accepted for publication once it meets all outstanding technical requirements.

Kind regards,

Academic Editor

PLOS ONE

Additional Editor Comments (optional):

Reviewers' comments:

Reviewer's Responses to Questions

**Comments to the Author**

1. Does the manuscript provide a valid rationale for the proposed study, with clearly identified and justified research questions?

Reviewer #1: Yes

Reviewer #3: Yes

2. Is the protocol technically sound and planned in a manner that will lead to a meaningful outcome and allow testing the stated hypotheses?

Reviewer #1: Yes

Reviewer #3: Yes

3. Is the methodology feasible and described in sufficient detail to allow the work to be replicable?

Reviewer #1: Yes

Reviewer #3: Yes

4. Have the authors described where all data underlying the findings will be made available when the study is complete?

Reviewer #1: Yes

Reviewer #3: Yes

5. Is the manuscript presented in an intelligible fashion and written in standard English?

Reviewer #1: Yes

Reviewer #3: Yes

6. Review Comments to the Author

You may also provide optional suggestions and comments to authors that they might find helpful in planning their study.

Reviewer #1: The authors have addressed all my comments/concerns. Congratulations on a well-written manuscript and great work!

Reviewer #3: The revised version of the manuscript was good written, designed and discussed and all reviewers comments have been addressed.

7. PLOS authors have the option to publish the peer review history of their article (what does this mean?). If published, this will include your full peer review and any attached files.

Reviewer #1: No

Reviewer #3: No

---

## [Editor Report · Acceptance letter]

26 Apr 2024

PONE-D-23-20961R2 

PLOS ONE

Dear Dr. McIntyre, 

I'm pleased to inform you that your manuscript has been deemed suitable for publication in PLOS ONE. Congratulations! Your manuscript is now being handed over to our production team.

Kind regards, 

on behalf of

Dr. Robert Jeenchen Chen 

Academic Editor

PLOS ONE